# A Ras-like domain in the light intermediate chain bridges the dynein motor to a cargo-binding region

**Courtney M Schroeder, Jonathan ML Ostrem, Nicholas T Hertz[†], Ronald D Vale***

Department of Cellular and Molecular Pharmacology, Howard Hughes Medical Institute, University of California, San Francisco, San Francisco, United States

**Abstract** Cytoplasmic dynein, a microtubule-based motor protein, transports many intracellular cargos by means of its light intermediate chain (LIC). In this study, we have determined the crystal structure of the conserved LIC domain, which binds the motor heavy chain, from a thermophilic fungus. We show that the LIC has a Ras-like fold with insertions that distinguish it from Ras and other previously described G proteins. Despite having a G protein fold, the fungal LIC has lost its ability to bind nucleotide, while the human LIC1 binds GDP preferentially over GTP. We show that the LIC G domain binds the dynein heavy chain using a conserved patch of aromatic residues, whereas the less conserved C-terminal domain binds several Rab effectors involved in membrane transport. These studies provide the first structural information and insight into the evolutionary origin of the LIC as well as revealing how this critical subunit connects the dynein motor to cargo.

***For correspondence:** vale@ ucsf.edu

**Present address:** [†]Laboratory of Brain Development and Repair, The Rockefeller University, New York, United States

**Competing interests:** The authors declare that no competing interests exist.

**Reviewing editor**: Anthony A Hyman, Max Planck Institute of Molecular Cell Biology and Genetics, Germany

## Introduction

Molecular motors transport a variety of cargos, including membranous organelles, proteins, mRNA, and chromosomes, along cytoskeletal tracks throughout the cell. Long distance transport in animal cells occurs primarily along microtubule tracks using kinesins (primarily, plus-end-directed) and cytoplasmic dyneins (minus-end-directed) as motors (*Allan, 2011*). The cytoplasmic dyneins are divided into two distinct subclasses. Cytoplasmic dynein 1 is employed broadly for many different types of retrograde microtubule transport within animal cells (*Allan, 2011*). Cytoplasmic dynein 2, in contrast, specifically carries out retrograde intraflagellar transport in cilia and flagella (*Ishikawa and Marshall, 2011*).

Cytoplasmic dynein is a large homodimer that consists of a heavy chain (>500 kDa) and several smaller associated subunits, each present in two copies. The dynein heavy chain includes an N-terminal elongated 'tail' domain followed by ~350 kDa motor domain that also contains the microtubule-binding domain (*Carter et al., 2011*; *Kon et al., 2012*; *Schmidt et al., 2012*). The heavy chain 'tail' domain binds the associated subunits, which include the intermediate chain (IC), the light intermediate chain (LIC), and the light chains (LC): Tctex1, LC8, and LC7/roadblock (*Allan, 2011*). A variety of studies have implicated these associated subunits in cargo binding, either directly, such as with rhodopsin (*Tai et al., 1999*), or indirectly by adaptors, such as dynactin (*Karki and Holzbaur, 1999*; *Schroer, 2004*).

The LIC subunits, which are present in all cytoplasmic dyneins described thus far but absent from axonemal dyneins (*Inaba, 2007*), are thought to play important roles in cargo transport. Invertebrates contain a single cytoplasmic dynein 1 LIC isoform while mammals have two LIC genes (LIC1 and LIC2) (*Hughes et al., 1995*), which may define two distinct cytoplasmic dynein 1 populations (*Tynan et al., 2000a*; *Tan et al., 2011*). Cytoplasmic dynein 2 is associated with a third LIC isoform (LIC3) (*Grissom et al., 2002*) that is required for retrograde intraflagellar transport (*Hou et al., 2004*). Knockdown studies have implicated LIC1 and 2 in membrane trafficking toward the endosomal-recycling compartment (ERC) in the cell center (*Horgan et al., 2010b*), ER export (*Kong et al., 2013*), lysosomal

**eLife digest** Living cells are constantly bustling with activity. They take in nutrients, carefully split their genetic information between new cells when they divide, and move their internal components into the right positions. To move these cargos around, the cell uses proteins—such as dynein—that essentially walks along the cell's internal scaffolding by making step-like movements. However, how a dynein motor protein is tethered to its cargo is not known in detail.

One part of the dynein structure thought to play an important role in binding the motor to its cargo is called the light intermediate chain (LIC). Schroeder et al. used X-ray crystallography to solve the structure of the light intermediate chain of dynein motors from a fungus. This information with other experimental techniques reveals that the LIC subunit has two distinct regions: one that binds to three different proteins that serve as adapters for cargo attachment, and one that binds to the rest of the dynein motor.

The structure of the LIC includes a fold that is also found in many proteins belonging to a family of enzymes called GTPases, suggesting that the LIC evolved from this family. GTPases use a molecule called GTP to release energy and often act as on–off switches for various processes inside cells. However, the fungal LIC subunit cannot bind to molecules called nucleotides—which can act as energy sources—the way GTPases do. This prevents the LIC subunit from acting as a molecular switch.

In contrast, the human version of the LIC is able to bind to some nucleotides, in particular one called GDP. However, since the LIC cannot bind to the high-energy nucleotide GTP, the human LICs most likely also do not act as on–off switches: Schroeder et al. instead propose that the LIC may use GDP only to stabilize the protein.

It remains to be seen how cargo attachment to the LIC is regulated. Further structural work and biochemistry with the LIC bound to the dynein motor and cargo will provide more insight into the mechanism of intracellular cargo transport.

localization and morphology (*Tan et al., 2011*), and axonal retrograde transport (*Koushika et al., 2004*). During mitosis, LIC1/2 are required for many cytoplasmic dynein 1 functions including centrosome anchoring, dynein localization to the kinetochore, progression through the spindle assembly checkpoint, and chromosome alignment (*Mische et al., 2008*; *Sivaram et al., 2009*; *Raaijmakers et al., 2013*). Several proteins implicated in membrane trafficking have been suggested to interact with LICs, including Rab4a (*Bielli et al., 2001*) and FIP3, a Rab11-family interacting protein (*Horgan et al., 2010a*, *2010b*). LIC1 and LIC2 appear to have largely redundant roles (*Kong et al., 2013*; *Raaijmakers et al., 2013*), although the centrosome protein pericentrin was reported to only bind to LIC1 (*Tynan et al., 2000a*) and Par3 binds specifically to LIC2 (*Schmoranzer et al., 2009*). In addition to mediating cargo binding, the LIC may be important for the stability of the dynein heavy chain; the LIC is the most stably bound subunit to the heavy chain (*King et al., 2002*) and is necessary for the stable expression and solubility of the recombinant human dynein heavy chain (*Trokter et al., 2012*).

Sequence comparisons show that the LICs are most conserved in the N-terminal half of the protein and that this conserved region contains a P-loop, a canonical nucleotide-binding sequence (*Perrone et al., 2003*). Based upon the high similarity to the P-loops of the ATP-hydrolyzing ABC transporters, it was suggested that the LIC may be an ATPase (*Hughes et al., 1995*). However, bioinformatic databases, such as Pfam (*Finn et al., 2014*), place the LIC in the same family as Ras-like, GTP-binding proteins. Several studies investigated the role of potential nucleotide hydrolysis by mutating a critical lysine residue in the P-loop but did not find a phenotype on cytoplasmic dynein function (*Tynan et al., 2000a*, *2000b*; *Yoder and Han, 2001*; *Hou et al., 2004*). However, beyond sequence analysis, little biochemical or structural information exists for the LIC subunits.

In this study, we report the crystal structure of the conserved N-terminal domain of the LIC from a thermophilic hyphal fungus, *Chaetomium thermophilum*, and show that it is composed of a canonical G protein fold. However, unlike most small GTP-binding proteins, the nucleotide pocket is empty, the P-loop exhibits a closed conformation and our biochemical experiments confirm that the fungal LIC G domain does not bind nucleotide. In contrast, we find that the human LIC1 G domain is capable of binding guanine nucleotides, particularly GDP. Our results reveal that the cytoplasmic dynein LIC

evolved from the small G protein superfamily and show biochemical differences between fungal and metazoan LICs that may play a role in dynein cargo transport. We further show how the LIC links the dynein motor domain to its multiple cargos.

## Results

### Crystal structure of a thermophilic yeast LIC domain

We originally attempted to crystallize full-length human LIC1 but were unable to obtain diffraction-quality crystals. Given prior success in crystallizing proteins from the thermophilic fungus *Chaetomium thermophilum* (*Amlacher et al., 2011*), we were inspired to crystallize the LIC from this organism. *C. thermophilum* possesses a gene that encodes a 4413 a.a. protein with ~50% and 30% sequence identity to the heavy chain of cytoplasmic dynein 1 from human and *Saccharomyces cerevisiae*, respectively. It also contains a LIC gene (EGS22626.1) that encodes for a 547-residue protein with 23% sequence identity and of similar size to human LIC1. For comparison, the *S. cerevisiae* LIC gene, which was originally identified based on a dynein-like nuclear migration phenotype (*Lee et al., 2005*), has 18% sequence identity with human LIC1. The sequence identity was higher with LICs from other hyphal fungi such as *Neurospora crassa* (75%) and *Aspergillus nidulans* (55%) (*Figure 1—figure supplement 1*). Sequence alignments show that the conservation is greatest in an N-terminal region of ~300 residues (predicted molecular weight of ~33 kDa) (*Figure 1A*; *Figure 1—figure supplement 1*).

The *C. thermophilum* LIC expressed well in *Escherichia coli* and could be purified to near homogeneity (see 'Materials and methods'). Crystals of the *C. thermophilum* LIC, which appeared after approximately 1 month, diffracted to 2.1 Å and a complete X-ray diffraction dataset was obtained (*Table 1*). However, these crystals were difficult to reproduce. When the crystals were analyzed by SDS-PAGE, three polypeptides corresponding to molecular weights of approximately 33, 27, and <15 kDa were observed but not the full-length 60 kDa LIC protein (*Figure 1B*). Mass spectrometry showed that the 33 kDa and 27 kDa polypeptides contained sequences that resided within the conserved N-terminal domain. These results suggested that the full-length LIC was being digested by a minor contaminating protease over the course of a month and that the protease-resistant region that crystallized corresponded to the most conserved region of the LIC (*Figure 1A*).

Due to the difficulty in reproducing the crystals, we were unable to obtain crystals with selenomethionine-labeled protein or heavy metal derivatives in order to obtain experimental phases. To produce the crystals with selenomethionine-labeled protein, we attempted in situ proteolysis during crystallization, as reported in prior studies (*Bai et al., 2007*). After testing a number of proteases, we found that chymotrypsin produced similar proteolysis products to those observed with our original crystals (*Figure 1B*). Next, we performed in situ proteolysis of selenomethionine-labeled *C. thermophilum* LIC in crystallization drops (molar ratio of 1:1000, chymotrypsin to LIC). Selenomethionine crystals appeared in 3 days and diffracted to 3.6 Å, and the subsequent dataset was phased by Multi-wavelength Anomalous Dispersion (MAD) (*Table 1*) and used to build an initial low-resolution model for the structure. This model was then successfully used in a molecular replacement search to phase the native 2.1 Å data set, previously obtained from our initial unlabeled protein crystals. After multiple rounds of refinement, we obtained a final 2.1 Å structure with an R-work of 17.4 and R-free of 22.0 (*Table 1*). Arg44 and Thr394 were the N-terminal and C-terminal residues that were visible in the electron density map; the molecular weight of the intervening polypeptide chain corresponds approximately to the highest molecular weight band (~33 kDa) seen from the crystals by SDS-PAGE (*Figure 1B*). However, the additional two polypeptide fragments seen by SDS-PAGE of the crystals (*Figure 1B*) suggest that internal cleavage also occurred, most likely in regions where electron density is missing from our structure (74–88, 202–210, and 346–374).

### The conserved LIC domain has a Ras-like fold

While the only sequence analysis in the literature suggested that the LIC might be an ATPase (*Hughes et al., 1995*), the Pfam sequence database classifies the LICs as belonging to the Ras-like, GTP-binding protein superfamily. The crystal structure of the conserved 33-kDa fragment of the LIC indeed supports this similarity with Ras. The LIC structure revealed a central β-sheet flanked by α-helices (*Figure 1C*), which in comparison with other protein structures in the Protein Data Bank (PDB) using the Dali server (*Holm and Rosenstrom, 2010*), is most similar to the G domain of small GTP-binding proteins (*Wittinghofer and Vetter, 2011*). Like Ras (*Pai et al., 1990*), the prototype of small G proteins,

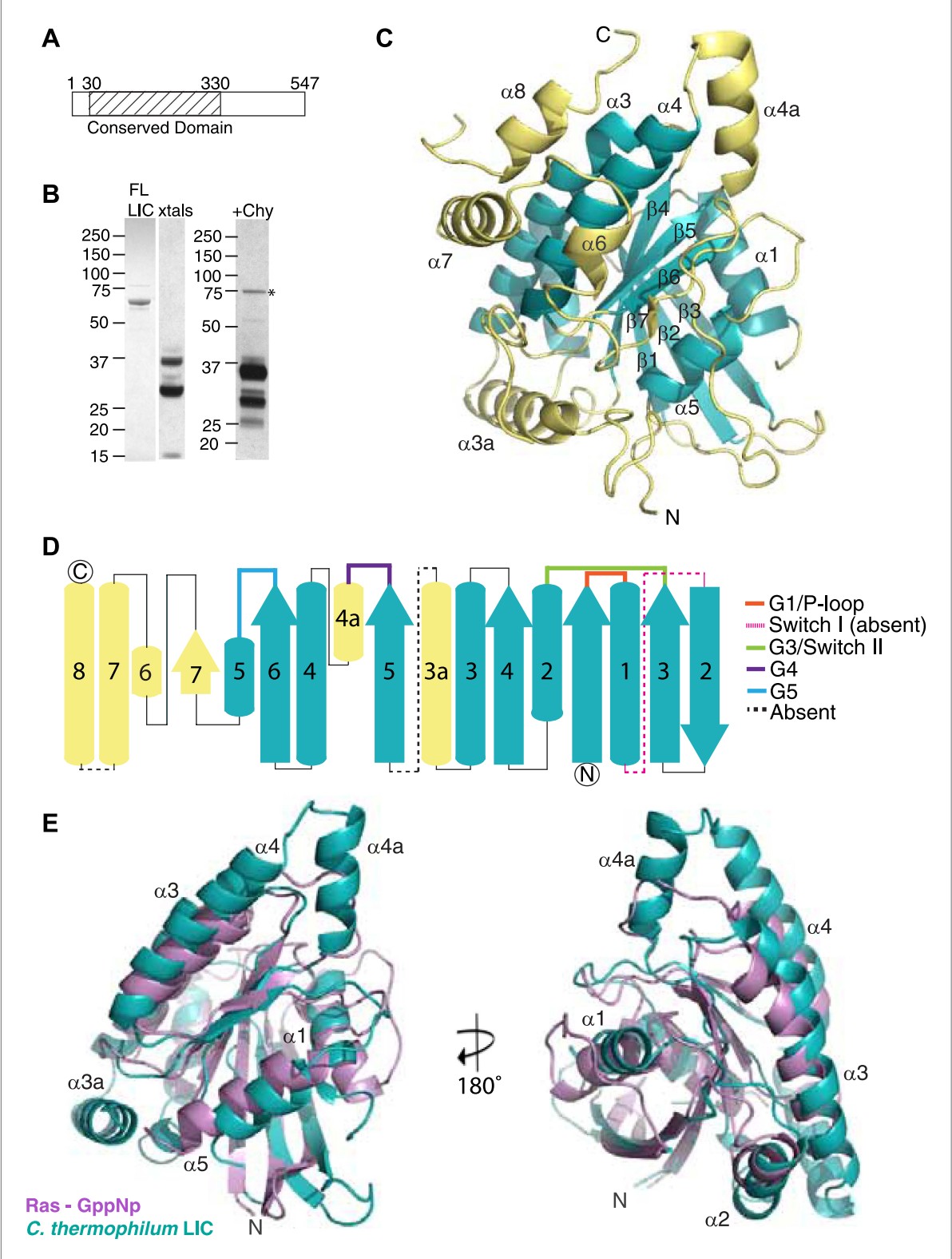

**Figure 1**. The dynein light intermediate chain has a Ras-like fold. (**A**) Diagram depicting the approximate range of conservation among all LICs (residue numbers with respect to the *C. thermophilum* LIC sequence). (**B**) The purified full-length *C. thermophilum* LIC (FL LIC) and the crystallized protein (xtals) were resolved on an SDS-PAGE gel and silver-stained, revealing proteolysis during crystallization. Proteolysis with chymotrypsin (+Chy) (overnight at 1:250 moles protease: LIC) produced similar sized fragments to those seen in the crystal. The asterisk marks a contaminating 75 kDa protein. (**C**) The

*Figure 1. Continued on next page*

*Figure 1. Continued*

2.1 Å structure of the *C. thermophilum* LIC is shown with the N-terminus oriented to the front and the C-terminus towards the back. β-strands and α-helices are labeled with respect to comparable elements in Ras. Elements that align with Ras are teal, and elements not found in Ras are yellow. (**D**) A topology map of LIC secondary structure is shown, and the color scheme corresponds to (**C**). Numbers with 'a' are additional inserts not seen in Ras. The P-loop, switch 1, switch 2, G4, and G5 motifs are labeled based on where they are found structurally (not based on sequence). Regions absent from the electron density are labeled with a dashed line. (**E**) Structural alignment of LIC with Ras-GMPPNP (PDB 52P1) (*Pai et al., 1990*). Alignment was performed using chimera after removing the C-terminal helices and loops in the LIC structure.

The following figure supplements are available for figure 1:

**Figure supplement 1**. Sequence alignment of full-length LICs.

**Figure supplement 2**. Structural and sequence similarity with the Rabs.

**Figure supplement 3**. Phylogenetic analysis of the LIC in the Ras superfamily.

LIC has six β-strands in the core of the structure (five parallel and one anti-parallel) as well as five similarly placed α-helices (*Figure 1C,E*). As a result of this structural similarity, we refer to this region as the LIC G domain. However, the LIC G domain also has conspicuous differences from Ras (*Figure 1C,E*). In addition to the five-α-helices found in Ras, the LIC G domain has two-α-helical insertions (α-helix 3a and 4a) that extend from the core of the structure (*Figure 1C–E*). Rab28 and Rab33 also have a similar α-helix 4a insertion, and the Dali server identifies these Rabs, followed by Rab32, as the most similar to the LIC structure (*Figure 1—figure supplement 2A*). The LIC G domain also has three additional α-helices at the C-terminus (α-helix 6, 7, and 8; *Figure 1C,D*) that interact with the core G domain structure (*Figure 1E*) and has a short seventh β-strand that extends the β-sheet (*Figure 1C,D*). These results indicate that the LIC is related to the Ras GTPase superfamily.

Despite structural similarity with G proteins, the LIC sequence has highly diverged from G proteins. Using a structure-based sequence alignment, the sequence similarity is only 12%, 8%, 8%, and 10% between the *C. thermophilum* LIC and Rab33, Rab28, Rab32, and H-Ras, respectively; the similar residues are mostly hydrophobic amino acids populating the central β-sheet (*Figure 1—figure supplement 2B*). We also performed a phylogenetic analysis of fungal and metazoan LICs within the Ras superfamily (*Rojas et al., 2012*). The phylogenetic tree, which reveals similar subfamilies as seen in past work (*Rojas et al., 2012*; *Basilico et al., 2014*), suggests that the LIC shares a common ancestor with the Ras/Rab subfamilies, whereas the SRPRBs and ARF families, basal subfamilies of the Ras superfamily, exclude the LIC clade (*Figure 1—figure supplement 3*).

## Thermophilic yeast LIC exhibits a closed nucleotide-binding pocket

G proteins have five unstructured loops with highly conserved residues that contribute to nucleotide-binding and hydrolysis (*Wittinghofer and Vetter, 2011*). These five motifs are G1 (or the P-loop, GxxxxGKS/T), G2 (or switch 1, a loop that includes a conserved threonine), G3 (or switch 2, DxxG), G4 (N/TKxD), and G5 (SAK). These canonical motifs are present in metazoan LIC1/2, but diverged considerably in fungal LICs, including *C. thermophilum*, suggesting that the pocket of fungal LICs might not bind the nucleotide (*Figure 2*). Consistent with this sequence information, our structure of the *C. thermophilum* LIC G domains shows a lack of electron density corresponding to a nucleotide in the binding pocket (*Figure 2—figure supplement 1*).

Remarkably, analysis of the *C. thermophilum* LIC binding pocket reveals significant structural deviations from Ras, which likely explain the absence of bound nucleotide. A similar glycine demarks the start of the P-loop at the end of β1. However, three residues (V57, D58, and S59) form an extra turn of α1 that shortens the 'P-loop' of *C. thermophilum* LIC compared to the canonical P-loop of Ras (*Figure 3A*). Unusual for G proteins, the 'P-loop' and 'switch 2' of the *C. thermophilum* LIC G domain interact with one another; the side chain of Q60 in the P-loop hydrogen bonds with the main chain carbonyl of G54 and the side chain of T116 in switch 2 (*Figure 3B*). This glutamine is found only in the P-loops of fungal LIC but is not present in metazoan LIC P-loops (*Figure 2*). As a result of this architecture, the *C. thermophilum* LIC P-loop occupies the space of the bound GTP analogue GMPPNP in Ras (PDB: 5P21) (*Pai et al., 1990*) and sterically clashes with the α- and β-phosphates (*Figure 3A*, inset).

**Table 1.** Crystallographic data and refinement statistics

| | Native | SeMet | |
|---|---|---|---|
| Data collection | | | |
| Space group | C 2 2 2$_1$ | P3$_2$21 | |
| Cell dimensions | | | |
| a, b, c (Å) | 59.37, 138.67, 112.54 | 58.81, 58.81, 198.23 | |
| α, β, γ (°) | 90, 90, 90 | 90.00, 90.00, 120.00 | |
| | | *Peak* | *Remote* |
| Wavelength | 1.115869 | 0.97973 | 0.95696 |
| Resolution (Å) | 50–2.10 (2.15–2.10)* | 50–3.50 (3.56–3.50)* | 50–3.50 (3.56–3.50)* |
| I/σI | 12.3 (1.7)* | 12.1 (1.6)* | 11.9 (1.5)* |
| Completeness (%) | 99.9 (99.7)* | 99.8 (96.8)* | 99.8 (98.0)* |
| Redundancy | 7.3 (7.4)* | 21.1 (11.9)* | 21.0 (12.0)* |
| †$R_{sym}$ | 0.18 (1.35)* | 0.23 (0.68)* | 0.24 (0.73)* |
| ‡$R_{pim}$ | 0.07 (0.48)* | 0.11 (0.22)* | 0.11 (0.23)* |
| CC$_{1/2}$ | 99.6 (53.3)* | | |
| Phasing | | | |
| Resolution | | 50–4.2 | |
| No. of SeMet sites | | 4 | |
| Initial figure of merit | | 0.32 | |
| Refinement | | | |
| Resolution (Å) | 50–2.10 | | |
| No. reflections | 27,513 | | |
| §$R_{work}$/$R_{free}$ | 17.4/22.0 | | |
| No. non-hydrogen atoms | | | |
| Protein | 2428 | | |
| Water | 122 | | |
| B-factors | | | |
| Protein | 35.6 | | |
| Water | 33.1 | | |
| R.m.s deviations | | | |
| Bond lengths (Å) | 0.012 | | |
| Bond angles (°) | 1.23 | | |
| Ramachandran favored (%) | 98.0 | | |
| Ramachandran outliers (%) | 0.0 | | |
| PDB code | 4W7G | | |

*Numbers in parentheses refer to the highest resolution shell.

†$R_{sym} = \sum_{hkl}\sum_i|I_{i(hkl)} - \langle I_{hkl}\rangle|/\sum_{hkl}\sum_i I_{i(hkl)}$, where $I_{i(hkl)}$ is the scaled intensity of the $i^{th}$ measurement of a reflection and $\langle I_{hkl}\rangle$ is the average intensity for that reflection.

‡$R_{pim} = \sum_{hkl} [1/(n-1)]^{1/2} \sum_i|I_{i(hkl)} - \langle I_{hkl}\rangle|/\sum_{hkl}\sum_i I_{i(hkl)}$, where $n$ is the number of times a single reflection has been observed.

§$R = \sum_{hkl}|F_{obs, hkl} - F_{calc, hkl}|/\sum_{hkl}|F_{obs,hkl}|\times 100$, where $R_{free}$ was calculated on a test set comprising approximately 6% of the data excluded from refinement.

These observations suggest that the P-loop and switch 2 loops are stabilized in a closed conformation that prevents nucleotide from binding to *C. thermophilum* LIC.

With regard to the other G motifs, density for the homologous G2 loop (switch 1) is absent in the LIC G domain structure, although this is also the case for many G proteins in a GDP-bound form

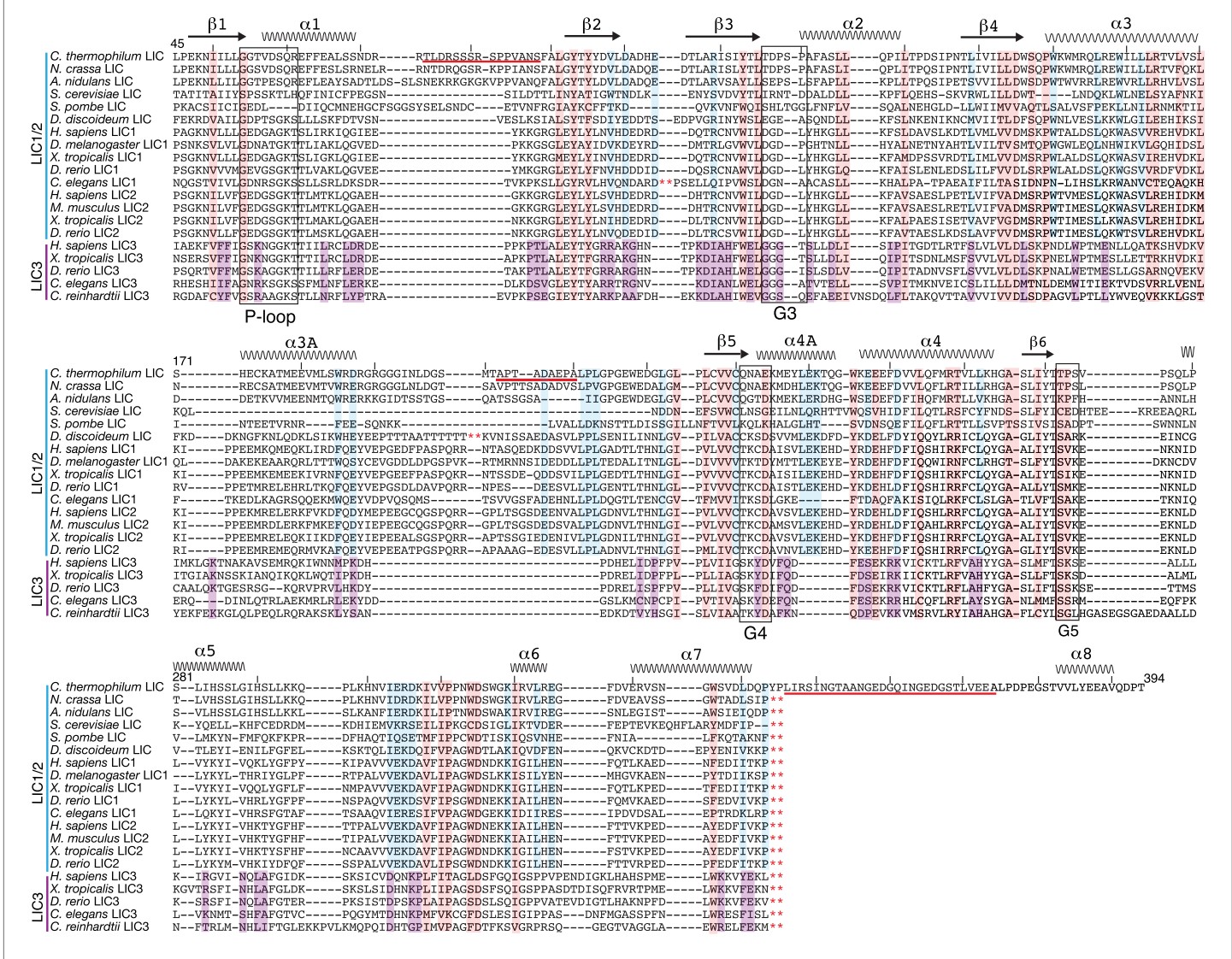

**Figure 2.** Conservation of LIC sequences and alignment with the LIC structure. 20 LICs were aligned via Promals (*Pei and Grishin, 2007*) (the *C. thermophilum* LIC pdb aided the alignment), and only the conserved sequence of the G domain is shown with the numbering based on *C. thermophilum* LIC. The P-loop, G3, G4, and G5 motifs were identified by the LIC structure and are labeled. The secondary structure of the *C. thermophilum* LIC is depicted above the sequences, and the residues missing in the structure are underlined in red. The red asterisks denote where sequence was taken out for space. Residues that were 80% conserved among only LIC1 and 2 sequences (12 out of 15), only LIC3s (4 out of 5), or universally conserved among all LICs (16 out of 20) are highlighted light blue, purple, and pink, respectively. Only the *C. thermophilum* LIC sequence extends to 394; all the other sequences were truncated with respect to *C. thermophilum* LIC, a.a. 343, because their predicted α-helix 8 extends beyond the alignment shown here.

The following figure supplement is available for figure 2:

**Figure supplement 1.** Electron density map of the P-loop and switch 2.

(*Wittinghofer and Vetter, 2011*). The G3 motif in G proteins follows after β3 and contains a conserved Asp that coordinates the magnesium ion and a glycine that hydrogen bonds with the γ-phosphate (DxxG) (*Wittinghofer and Vetter, 2011*). However, the same loop in the *C. thermophilum* LIC G domain lacks an Asp and a Gly in a comparable position. The loops corresponding to the G4 motif and G5 motif are in similar positions in the *C. thermophilum* LIC G domain as in Ras (*Figure 3A*). Collectively, the structural data suggest that the G domain of the fungal LICs have lost their ability to bind the nucleotides.

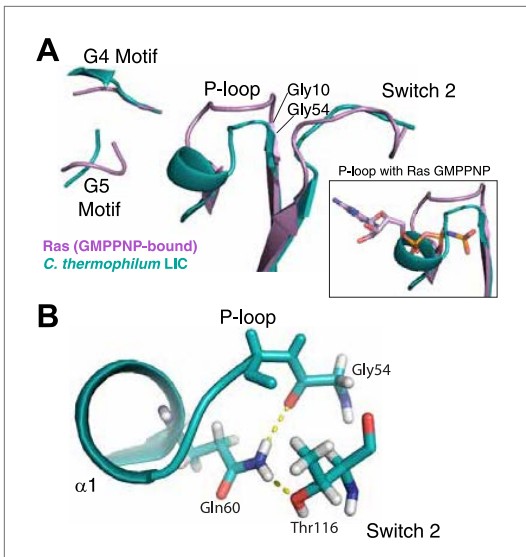

**Figure 3**. The LIC G domain binding pocket exhibits a closed conformation that is not compatible with nucleotide binding. (**A**) The *C. thermophilum* LIC was aligned with Ras-GMPPNP (PDB 5P21) and a view of the GTP-binding pocket is shown with corresponding G motifs labeled. GMPPNP is not shown. The inset shows the aligned P-loops with Ras in complex with GMPPNP. (**B**) Interactions between the P-loop and switch 2 of *C. thermophilum* LIC are shown with a dashed yellow line.

## Human, but not *C. thermophilum*, LIC binds guanine nucleotides

The sequence and structural data described above suggest that the *C. thermophilum* LIC G domain may not be able to bind nucleotide while metazoan LICs, which contain a more canonical P-loop motif, may have nucleotide-binding capability. To test these ideas, we directly assayed for nucleotide-binding by *C. thermophilum* LIC and human LIC1 G domains purified after *E. coli* expression (see 'Materials and methods'). When the *C. thermophilum* LIC G domain was injected onto a C18 column for reverse-phase liquid chromatography (RPLC) with acetonitrile, no 260 nm adsorbing molecule eluted from the column (*Figure 4A*), which is consistent with the lack of nucleotide in the crystal structure. In contrast, when the same experiment was performed with the human LIC1 G domain, a small molecule eluted with a retention time that did not match either GDP or GTP but which had an adsorption spectrum characteristic of a guanine nucleotide (*Figure 4B,C*; *Figure 4— figure supplement 1A*). The buffer used in the RPLC included tetrabutylammonium hydroxide, which binds to negatively charged entities, making them hydrophobic. Since the column retains hydrophobic molecules longer, more negatively charged molecules exhibit longer retention times. The molecule extracted from the LIC1 G domain has a longer retention time than even guanosine tetraphosphate, suggesting that it is slightly more negatively charged (*Figure 4B*). When the small molecule was separated from the LIC1 G domain before injecting it on the C18 column, it had the same retention time (*Figure 4—figure supplement 1B*), suggesting that the long retention time was not an artifact of simultaneously injecting it along with protein on the column.

A guanine nucleotide that is more charged than guanosine tetraphosphate is guanosine-3',5'-bisdiphosphate (termed ppGpp), which has two diphosphates connected to the 5' and 3' hydroxyls of ribose (*Figure 4D*). ppGpp, which is produced as part of the stringent response in *E. coli* (*Chatterji and Ojha, 2001*), has been co-crystallized with the bacterially expressed GTPases Arl5 (bound to a likely ppGpp remnant GDP3'P; PDB: 1ZJ6) (*Wang et al., 2005*) and Obg (PDB: 1LNZ) (*Buglino et al., 2002*). Within the margin of variation of different chromatographic runs (*Figure 4—figure supplement 1C*), the retention time of 150 μM ppGpp was similar to that of the guanine nucleotide co-purifying with 150 μM human LIC1 (*Figure 4D*; *Figure 4—figure supplement 1C*). To determine if the nucleotides were truly identical, we combined and injected 150 μM ppGpp and 150 μM human LIC1 G domain. Only one peak was detected, which corresponded to the sum of the absorbances of the commercial ppGpp and the nucleotide co-purifying with the human LIC1 G domain injected separately (*Figure 4D*). Thus, the nucleotide in the human LIC1 G domain co-purifies precisely with ppGpp in the same chromatographic run. The eluted human LIC1 nucleotide was not successfully identified by mass spectrometry, most likely due to its high negative charge and low quantity following extraction, and thus additional information on its identity could not be obtained by this method. However, with indistinguishable RPLC elution times and a guanine-like absorption spectrum, we believe that ppGpp is the most likely candidate for the LIC-bound nucleotide.

The nucleotide ppGpp can reach millimolar concentrations under bacterial stress (*Srivatsan and Wang, 2008*), as can occur during the growth conditions for protein overexpression. However, ppGpp is thought to only exist in bacteria and plants and not human cells (*Takahashi et al., 2004*; *Srivatsan and Wang, 2008*). Thus, we do not believe that ppGpp is a natural ligand for the human LIC, but that

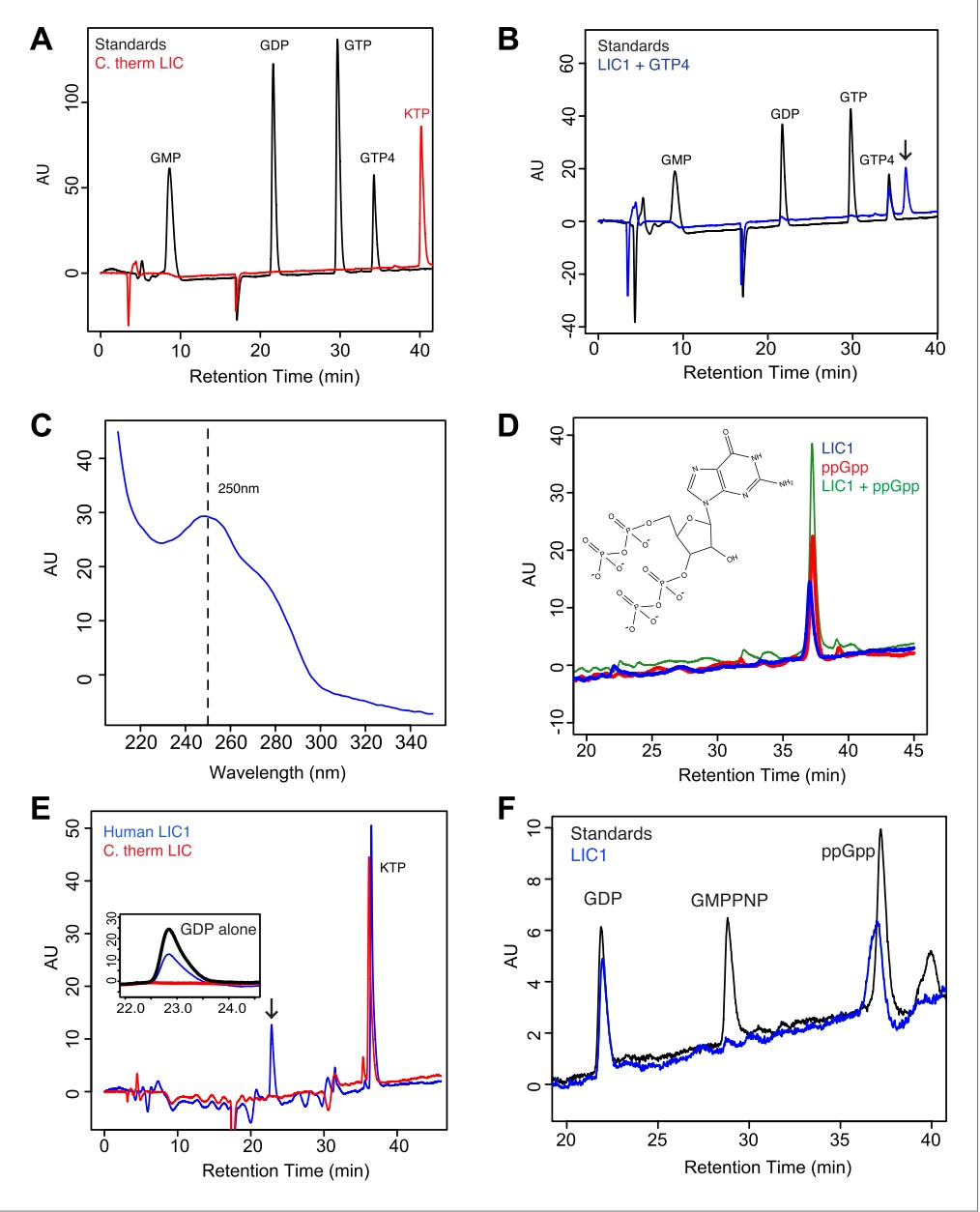

**Figure 4**. The human LIC1 G domain binds guanine nucleotide. (**A**) The *C. thermophilum* LIC G domain (a.a. 45–394) was injected on the C18 column with an increasing gradient of acetonitrile and detection with a wavelength of 260 nm. Standards are at 1 mM, and kinetin triphosphate (KTP), a non-biological nucleotide (*Hertz et al., 2013*), was added for an internal control. KTP was used as a positive control for sample injection because it elutes after all other nucleotides due to its high negative charge. (**B**) The human LIC1 G domain (amino acids 65–354) was analyzed by RPLC as done in (**A**). Nucleotide standards are at 0.5 mM, and the LIC was simultaneously injected with an equal concentration of guanosine tetraphosphate (GTP4) for an internal reference. An arrow indicates LIC nucleotide. (**C**) The wavelength spectrum of the LIC nucleotide in (**B**, arrow) is shown from 210 nm to 350 nm. (**D**) The human LIC1 G domain at 150 μM and ppGpp at 150 μM were analyzed by RPLC separately. LIC1 and ppGpp, each at 150 μM were then injected simultaneously. The structure of guanosine-3′,5′-bisdiphosphate (ppGpp) is shown. (**E**) The human LIC1 G domain and *C. thermophilum* LIC G domain were incubated with 5 mM EDTA and 1 mM GDP for 1 hr at room temperature. An excess of MgCl₂ was then added at a final concentration of 10 mM and the resulting protein, with KTP as an internal control, was analyzed by RPLC. The inset shows the GDP standard alone superimposed with the two chromatograms. (**F**) The human LIC1 G domain (150 μM) was
*Figure 4. Continued on next page*

*Figure 4. Continued*

incubated with 5 mM EDTA, 0.5 mM GDP, and 5 mM GMPPNP for 1 hr at room temperature and analyzed by RPLC as done in (**E**).

The following figure supplements are available for figure 4:

**Figure supplement 1**. Guanine nucleotide extraction from human LIC1.

**Figure supplement 2**. Nucleotide exchanges with GTP and ADP.

**Figure supplement 3**. Human LIC1 co-purifies with GDP from human cells.

**Figure supplement 4**. Instability of human LIC1 G domain without nucleotide.

binding of this ligand reflects an unnatural situation of bacterial expression. Nevertheless, ppGpp binding indicates that the metazoan LIC is indeed competent to bind nucleotide and also raised questions of its nucleotide specificity, especially since we did not detect bound GTP, which is present in millimolar concentrations in bacteria (*Buckstein et al., 2008*). Thus, we hypothesized that human LIC1 might have a higher affinity for guanosine diphosphates than triphosphates. As was observed in the crystal structure of Arl5 (*Wang et al., 2005*), LIC1 might bind only one of the two diphosphates of ppGpp, which can be more abundant than GDP during bacterial stress (*Srivatsan and Wang, 2008*). To test whether the human LIC1 G domain can indeed also bind GDP, the protein was incubated with 1 mM GDP and 5 mM EDTA (to promote the release of bound ppGpp). After quenching the exchange reaction with 10 mM $MgCl_2$ and removing the unbound nucleotide, the protein was subjected to RPLC to analyze the composition of the bound nucleotide. Two peaks were now detected corresponding to the retention times of GDP and ppGpp, indicating that GDP in solution had partially exchanged for the ppGpp bound to the protein after bacterial expression (*Figure 4E*). In a similar nucleotide exchange experiment, we found that ADP did not bind to human LIC1 indicating that this G domain is indeed guanine-specific (*Figure 4—figure supplement 2A*). Furthermore, when the identical GDP exchange was performed with the *C. thermophilum* LIC G domain, the GDP in solution did not bind to the fungal protein (*Figure 4E*). Therefore, the human LIC1 G domain, but not *C. thermophilum* LIC, is capable of binding GDP and is most likely binding one diphosphate of ppGpp, as found previously with Arl5 (*Wang et al., 2005*). Note, approximately 2.5 mg/ml of LIC1 was required to detect nucleotide above the noise in the described RPLC method; therefore, it is possible that GDP also may co-purify with bacterially expressed human LIC1, but its abundance is below our detection limit.

We next examined relative affinity of the LIC G domain for guanosine diphosphates versus triphosphates. We incubated 150 μM human LIC1 G domain with 5 mM EDTA and a 10-fold excess of GMPPNP (5 mM) to GDP (0.5 mM GDP); the non-hydrolyzable GMPPNP was used to prevent any possibility of binding and conversion to GDP by enzymatic hydrolysis. GMPPNP binds to many G proteins and has been widely used as a non-hydrolyzable analogue (*Wittinghofer and Vetter, 2011*). After quenching the exchange reaction with 10 mM $MgCl_2$, the protein was analyzed by RPLC. The eluted material contained a mixture of GDP and non-exchanged ppGpp but no GMPPNP was detected despite its 10-fold excess to GDP in the starting reaction (*Figure 4F*). We also repeated the same exchange reaction with hydrolyzable GTP (again in 10-fold excess to GDP) but did not detect any GTP bound to human LIC (*Figure 4—figure supplement 2B*). Thus, these experiments reveal that the human LIC1 G domain has a higher affinity for guanosine diphosphates than guanosine triphosphates.

The nucleotide exchange experiments suggested that human LIC1 most likely is bound to GDP and not GTP in its native environment. To test this further, we expressed the human LIC1 G domain in HEK-293T cells, a human cell line. After purifying the protein, any potentially bound nucleotide was released by boiling. LC-MS analysis of the non-proteinaceous supernatant clearly revealed the presence of GDP, but no detectable GTP (*Figure 4—figure supplement 3*). Porcine brain tubulin was used as a positive control to demonstrate that the LC-MS could detect both GDP and GTP. Thus, human LIC1 expressed in human cells co-purifies with GDP alone. Thus, our results collectively suggest that metazoan LIC preferentially binds guanosine nucleotides containing a diphosphate (either GDP or ppGpp) that extends into the binding pocket.

## Regions of conservation in the LIC G domain

We next sought to examine the positions of conserved residues on the surface of the LIC G domain, which might be candidate residues that participate in binding to other proteins (*Figure 5A*). We classified residues as (1) conserved throughout all LICs, (2) LIC1/2-specific, or (3) LIC3-specific (considered conserved if 80% similar in each category) (*Figure 2*). Relatively few universally conserved residues are on the surface of the LIC G domain, with the exception of a hydrophobic groove of universally conserved residues composed of three aromatic residues (Y93, Y95 [β2], and Y113 [β3]) and Leu125 (α2) (residue numbers refer to the position in *C. thermophilum* LIC) (*Figure 5A*). A group of highly conserved, class-specific LIC3 residues surround this universally conserved, hydrophobic patch (*Figure 5A*). In LIC1/2, the corresponding residues are often reversed in charge or neutralized. A patch of LIC1/2-specific acidic residues (including E325, E304, and D306) lie on the opposite side of the molecule that includes the C-terminal loops and α-helix 4, 4a, and 6 (*Figure 5A*). Three LIC3-specific patches also lie in this same region (*Figure 5A*).

The sequence alignment in *Figure 2* also revealed unique insertions or deletions in different LIC species or isoforms. Hyphal fungi have a longer loop between α1 and β2. The LIC3s have a longer loop between α6 and α7 and a shorter loop between α3A and β5 than other LICs.

## Interaction of the LIC with the dynein heavy chain

We next sought to examine which domain of the LIC interacts with the dynein heavy chain by expressing full-length or fragments of HA-tagged human LIC1 in HEK-293T cells, immunoprecipitating the expressed LIC fragment, and probing for the dynein heavy chain by immunoblot analysis. The fragments tested were an N-terminal fragment of the LIC G domain (a.a. 1–389, which terminates after a predicted helix that may correspond to α8) and a C-terminal fragment (a.a. 389–523). We found that the N-terminal fragment, a.a. 1–389, bound the heavy chain with a comparable efficiency to the full-length human LIC but the C-terminal domain did not bind (*Figure 5B*). We also tested a longer C-terminal fragment starting after α7 (a.a. 355–523) and also saw no binding (data not shown). These results suggest that the heavy chain-binding interface lies within the highly conserved G domain and not in the C-terminal domain.

Our sequence analysis identified conserved solvent-exposed residues on the surface of the G domain (*Figure 5A*), and we wished to determine if any of these regions were involved in heavy chain binding. Double or triple alanine mutations were made in full-length human LIC1 targeting LIC1/2-specific regions (1: D110A/D112A/D113A; 2: V316A/E317A/D319A; 3: E317A/K332A; 4: R260A/E262A/E338A) and the one patch of universal conservation (5: Y101A/Y103A; 6: Y101A/Y103A/W120A). The mutant LIC1s were then expressed in HEK-293T cells and tested for heavy chain binding by immunoprecipitation and immunoblot analysis. Strikingly, the mutations of the universally conserved aromatic residues exhibited dramatically decreased binding to the heavy chain; the double mutant Y101A/Y103A exhibited decreased binding to the dynein heavy chain, and the binding appeared to be almost entirely disrupted in the triple mutant Y101A/Y103A/W120A (*Figure 5C*). The inability of these mutants to incorporate into the dynein holoenzyme complex was also verified by probing for the dynein intermediate chain (*Figure 5C*). In contrast, the other five double or triple mutations tested bound normally to the dynein heavy chain. These results indicate that the universally conserved hydrophobic groove on the LIC is involved in binding to the dynein heavy chain.

## Interaction of the LIC with cargo adapter proteins

We next investigated how the LIC binds to proteins that link dynein to Rabs involved in membrane transport. FIP3 and RILP are both adapter proteins that have been shown previously to bind both to mammalian LIC (*Horgan et al., 2010a*; *Horgan et al., 2010b*; *Scherer et al., 2014*) and Rab GTPases, Rab11 in the case of FIP3 (*Hales et al., 2001*) and Rab7 in the case of RILP (*Cantalupo et al., 2001*). LIC1 has also been implicated in dynein's interaction with another Rab effector—BicD2 (*Splinter et al., 2012*), an effector of Rab6 (*Matanis et al., 2002*), although whether this occurs through a direct interaction or not is unknown. We purified the LIC1 G domain (a.a. 1–389), the C-terminal domain (a.a. 389–523), and the full-length human LIC1 from bacteria to determine the binding site for cargo in vitro. The GST-tagged domains and full-length LIC1 were bound to glutathione beads and incubated with recombinant GFP-tagged FIP3, RILP, or BicD2. Binding was determined by centrifuging the beads and immunoblotting with an anti-GFP antibody. All three Rab effectors bound to the full-length LIC1 but not to untreated beads. This result also clearly establishes that BicD2 can bind directly to the LIC.

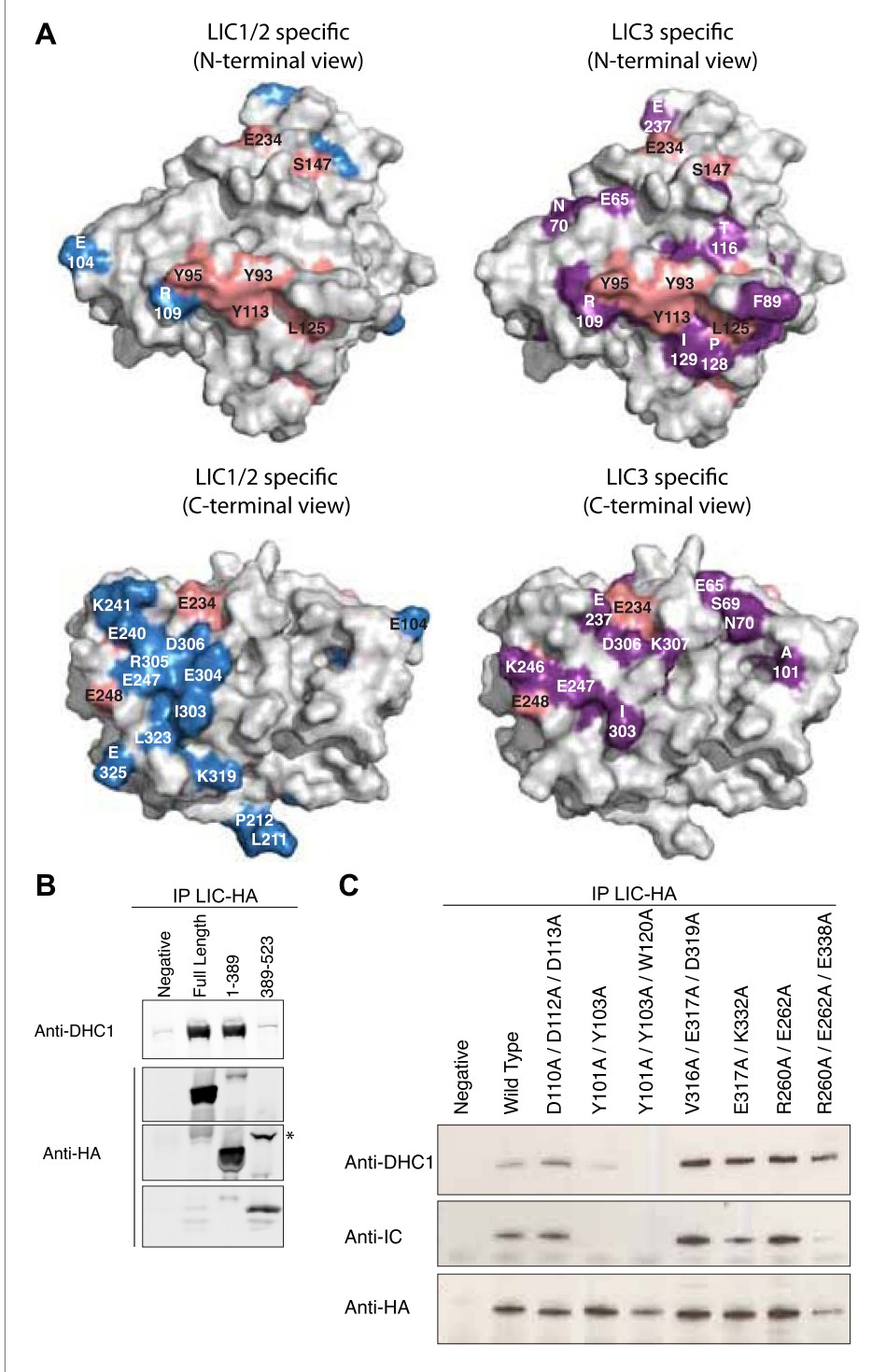

**Figure 5**. The G domain contains the binding interface of the dynein heavy chain. (**A**) The conservation of residues shown in *Figure 2* is mapped onto the surface of the *C. thermophilum* LIC G domain. The surface is shown in two different orientations with each orientation showing universally conserved residues (pink) and LIC1/2-specific residues (blue) vs LIC3-specific residues (purple). The LIC is oriented either toward the N-terminus or toward the C-terminal loops (as in *Figure 1C*). Conserved amino acids and the corresponding residue numbers are labeled according to the *C. thermophilum* LIC sequence. (**B**) HA-tagged fragments of human LIC1 (a.a. 1–389, 389–523) were expressed in HEK-293T cells, immunoprecipitated with an anti-HA antibody, and immunoblotted for the
*Figure 5. Continued on next page*

*Figure 5. Continued*

dynein heavy chain or HA tag. The asterisk denotes a non-specific band that reacts with the anti-HA antibody.
(**C**) HA-tagged double and triple mutants of human LIC1 were expressed, immunoprecipitated, and analyzed as in
(**B**) with additional immunoblotting for the dynein intermediate chain. The residue numbers shown correspond to
human LIC1. *Homo sapiens* (*H.s.*) LIC1 residues correspond to *C. thermophilum* (*C.t.*) LIC as follows: *H.s.* D110,
D112, D113 = *C.t.* D102, E104, D105; *H.s.* Y101, Y103, W120 = *C.t.* Y93, Y95, Y113; *H.s.* V316A, E317A, D319A = *C.t.*
I303, E304, D306; *H.s.* E317A, K332A = *C.t.* E304, K319; *H.s.* R260A, E262A, E338A = *C.t.* E325, E248, K246.

Interestingly, FIP3, RILP, and BicD2 each bound the C-terminal domain almost as equally well as to
full-length LIC1 but showed no interaction with the G domain (*Figure 6A–C*). The assay was also
conducted with GFP alone, which showed no binding to the full-length LIC1 or its truncations
(*Figure 6—figure supplement 1*). Thus, while the G domain contains the primary binding site for the
dynein heavy chain, the LIC C-terminal domain appears to serve as a docking site for cargo adaptors.

## Discussion

In this study, we provide the first structural information on a key cargo binding subunit of cytoplasmic
dynein, the light intermediate chain. We show that the dynein LIC contains a conserved domain with a
G protein fold. However, in comparison to Ras, the LIC domain contains two helical insertions in the
core structure, a short extra β-strand and additional C-terminal helices that pack against the core,
which collectively define a unique G domain topology that sets the LIC apart from previously described
G protein families (*Wittinghofer and Vetter, 2011*). The fungal LICs further differentiate themselves
from other small G proteins by their closed P-loops and inability to bind nucleotide, while the meta-
zoan LICs also show an unusual strong preference for binding GDP over GTP. Interestingly, the
C-terminal half of the protein is involved in binding Rab effectors, whereas the G domain is the docking
site for the dynein heavy chain. Collectively, these results provide insights into the evolutionary origin
of the LIC and how this subunit participates in dynein functions.

### Nucleotide's role in LIC function

The evidence that human LIC1 can bind nucleotide, unlike its fungal homologues, raises the question
as to whether or not metazoan LIC acts as a GTPase switch. Most small GTP-binding proteins change
their binding affinity for protein partners between GTP and GDP states, and one could envision a sim-
ilar role for a nucleotide switch in cargo selection and binding by metazoan LICs. However, our data
suggest that LICs do not function as GTPase switches. One reason is that the canonical GTPase switch
2 motif (DxxG), which plays a critical role in GTP hydrolysis (*Wittinghofer and Vetter, 2011*), appears
to be absent from the equivalent loop in metazoan LICs (*Figure 2*). Furthermore, human LIC1 binds
guanosine diphosphate preferentially over guanosine triphosphate; even when GMPPNP or GTP is in
10-fold excess over GDP, little or no GTP binding to the LIC is detected (*Figure 4F*). This preference
for guanosine diphosphates explains why human LIC1 co-purifies with ppGpp from bacteria, even
though the GTP concentration is approximately 1.6 mM in *E. coli* during exponential growth (*Buckstein
et al., 2008*). In contrast, the well-characterized GTPase Ras has a 10-fold higher affinity for GTP than
GDP (*John et al., 1990*). There are examples of G proteins that prefer guanosine diphosphates, such
as ADP-ribosylation factor (ARF), which has a >25-fold higher affinity for GDP than GTPγS (*Weiss
et al., 1989*). ARF can exchange GTP for GDP under particular conditions (*Weiss et al., 1989*), and it
is possible that metazoan LICs can do so as well. However, given the combination of preferential
GDP binding and the lack of a canonical switch 2 sequence, we find it unlikely that the LIC is a true
GTPase. While metazoan LIC may not switch between a GDP-bound and a GTP-bound state, it is pos-
sible that it reversibly exchanges GDP and adopts an apo form. Small GTPases are stable in the apo
form only when bound to a guanine-nucleotide-exchange factor, or GEF (*Cherfils and Chardin, 1999*);
by analogy, LIC1 may release GDP upon binding the dynein heavy chain. This idea could be tested in
the future by examining the nucleotide state of the LIC when in complex with the dynein heavy chain.

Another hypothesis for the role of nucleotide binding by metazoan LIC is simply to provide struc-
tural stability. In support of a role of nucleotide in structural stability, we have found that human LIC1
is unstable with point mutations that interfere with nucleotide binding (K80A in the P-loop or D248A
in the G4 motif; *Figure 4—figure supplement 4*). Surprisingly, overexpression of the P-loop K80A
mutation in *Caenorhabditis elegans* rescued the null LIC1 phenotype (*Yoder and Han, 2001*), but

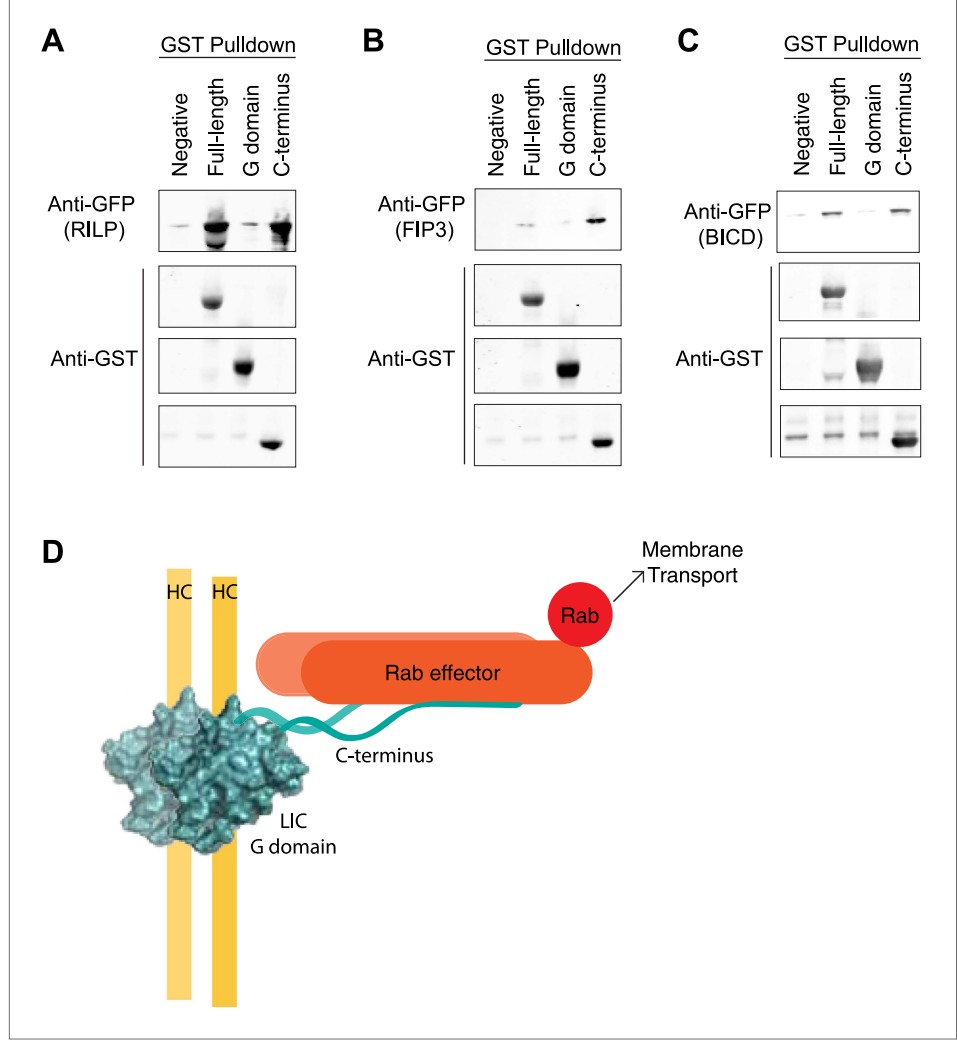

**Figure 6**. The LIC C-terminus alone binds Rab effectors. (**A**) Human GST-tagged full-length LIC and LIC truncations, including the G domain (a.a. 1–389) and the C-terminus (a.a. 389–523), were purified from *E. coli* bound to glutathione beads and incubated with recombinant GFP-FIP3. The beads were centrifuged, washed, and probed with an anti-GFP antibody to assess binding of the GFP-tagged protein. (**B**) The same experiment in (**A**) was done with GFP-RILP. (**C**) The same experiment in (**A**) was done with GFP-BicD2. (**D**) A model depicts the LIC G domain bound to the dynein heavy chain (HC) with the C-terminus bound to a Rab effector, allowing for membrane transport by dynein.
The following figure supplements are available for figure 6:

**Figure supplement 1**. Controls for GST pulldowns.

**Figure supplement 2**. Disorder probability of human LIC1.

endogenous expression levels may reveal a phenotype associated with instability of the mutant protein. The fungal LICs appear to have evolved another strategy for achieving protein stability in the absence of bound nucleotide. Two residues, Gln60 and Thr116, in the *C. thermophilum* LIC appear to play an important role in stabilizing a nucleotide-free state and preventing nucleotide binding (*Figure 3B*). Gln60 replaces the P-loop lysine, which is critical for nucleotide binding and hydrolysis, and Thr116 replaces the aspartate found in switch 2 (DxxG), which contacts the nucleotide-bound $Mg^{2+}$ ion via a water molecule (*Wittinghofer and Vetter, 2011*).

'Pseudo-ATPase' is a recently coined term for proteins with a structure typical of ATPases but which does not catalyze ATP hydrolysis and uses nucleotide for protein stability (*Adrain and Freeman, 2012*).

For example, the kinetochore nucleotide-binding protein BUBR1 does not hydrolyze ATP in carrying out its function in the mitotic checkpoint complex but rather requires ATP for its stability (*Suijkerbuijk et al., 2012*). We propose that metazoan LIC may be a 'pseudo-GTPase', where nucleotide is used for stability rather than in a cycle of conformational change. Another protein that also has a common G-protein fold with no nucleotide-binding or hydrolysis capability is CheY, a bacterial chemotactic protein (*Artymiuk et al., 1990*; *Chen et al., 1990*). A recent study also identified the kinetochore protein, CENP-M, as a 'pseudo-GTPase', since it contains a G protein fold but does not contain bound nucleotide (*Basilico et al., 2014*). Phylogenetic analysis suggests that the three identified 'pseudo-GTPases', LIC, CENP-M, and CheY, are all most similar to the Rab and Ras subfamilies of the G protein superfamily (*Figure 1—figure supplement 3*) (*Basilico et al., 2014*).

### Interactions of the LIC with the dynein motor and cargos

In this study, we have shown that the G domain alone binds the heavy chain and identified a groove of highly conserved, aromatic residues that are involved in this interaction. However, since this hydrophobic patch is conserved among all LICs, other residues must determine the targeting of LIC1/2 to cytoplasmic dynein 1 and LIC3 to cytoplasmic dynein 2. Our structure, combined with sequence analysis, provides some clues on residues that might be involved in the selective recognition of a particular dynein heavy chain. For example, the LIC3s have several class-conserved and mostly hydrophobic residues (conserved among LIC3s but not LIC1/2) that surround the hydrophobic groove (*Figure 5A*). LIC1/2 exhibits fewer class-conserved residues on the face of the G domain with the hydrophobic groove, although a few such residues exist (*Figure 5A*). Future structure–function binding studies with cytoplasmic dynein 1 and 2 and LIC1 and LIC3 will be needed to fully understand the nature of this selective interaction.

Our studies show that the LIC's C-terminal domain is involved in binding three different cargo adapter proteins, FIP3, RILP, and BicD2, and thus appears to contain multiple recognition sites for distinct adapter proteins. Interestingly, the C-terminal domain is predicted to be disordered (*Figure 6—figure supplement 2*). The LIC domain structure has interesting parallels with the dynein intermediate chain, which has a well-ordered WD repeat domain that interacts with the heavy chain and a less conserved disordered region that binds the dynein light chains (*Mok et al., 2001*; *Nyarko et al., 2004*) and cargo-adaptors like dynactin (*Vaughan and Vallee, 1995*). A number of disordered proteins become ordered upon binding other proteins. It will be interesting to determine what structure the C-terminal tail of the LIC adopts before and after binding to Rab effectors. While this study points to the C-terminal tail of LIC as a cargo binding site, it is possible that adapter proteins may be discovered in the future that bind to the G domain, perhaps via LIC1/2 specific residues.

Overall, our data suggest a working model in which the LIC domains are specialized for distinct functions. The G domain is docked onto the dynein heavy chain, while the C-terminal domain binds many of the LIC's protein partners, specifically Rab effectors involved in membrane transport (*Figure 6D*). Recent studies have shown that these Rab effectors, in addition to linking dynein to membrane cargos, link dynein to dynactin to create an ultraprocessive motor (*McKenney et al., 2014*; *Schlager et al., 2014*). Thus, the LIC may have a critical role in the mechanism of generating an ultraprocessive dynein–dynactin complex. While the LIC may simply serve as a docking site for Rab effectors, it is also possible that the LIC N- and C-terminal domains communicate allosterically in some way with other dynein chains and dynactin to affect their activities.

## Materials and methods

### Protein purification and crystallization

The homologous dynein LIC in *C. thermophilum* (EGS22626.1) was PCR-amplified from *C. thermophilum* cDNA, generously given by Peter Walter's lab at UCSF. The full-length protein was cloned into the vector pGEX-6P-1, which encodes an N-terminal GST tag. The protein was expressed in BL21 DE3 RIPL cells and induced with 0.5 mM IPTG for 3 hr at 37°C. The protein was purified using glutathione agarose 4B (USB) and then cleaved with GST-tagged human rhinovirus 3C protease overnight at 4°C. The cleaved product was further purified by gel filtration into 10 mM Tris–HCl (pH 7) and 25 mM NaCl using a HiPrep 16/60 Superdex S-200 HR column (GE Healthcare, Piscataway, NJ) with an AKTA FPLC system (GE Healthcare). The full-length protein eluted as a monomer, which was verified by static light scattering. Crystal trials were setup with 16.6 mg/ml protein by hanging drop vapor diffusion. Native

crystals were obtained at 20°C in a condition including 0.1 M MES (pH 6.5) and 20% (wt/vol) PEG 10000 in a Nextal PEGs screen (Qiagen Inc., Valencia, CA). The crystals were cryoprotected by the addition of 18% glycerol to the well solution and flash-cooled by plunging in liquid nitrogen.

To express selenomethione-labeled *C. thermophilum* LIC, protein was expressed in M9 minimal media. Before induction, the culture was incubated for 30 min with a cocktail including lysine, phenyl-alanine, threonine, isoleucine, leucine, valine, and selenomethione to inhibit methionine biosynthesis. The culture was then induced with 0.5 mM IPTG for 3 hr at 37°C. The labeled protein was purified in the same way as native protein except with the addition of 5 mM TCEP in the gel filtration buffer. Chymotrypsin (Sigma Aldrich, St. Louis, MO) was added to the protein immediately before setting up crystal trays at a ratio of 1:1000 moles (chymotrypsin to LIC) with the LIC at 8.6 mg/ml. Crystals were obtained at 20°C from a condition including 0.2 M calcium acetate, 0.1 M Tris (pH 7), and 20% (wt/vol) PEG 3000 (Qiagen). The crystals were cryoprotected by the addition of 18% glycerol to the well solution and flash-cooled by plunging in liquid nitrogen.

For bacterial expression, strep-tagged human LIC1 (NM_016141.3) and its truncations were cloned into pGEX-6P-1, which encodes an N-terminal GST tag. For mammalian cell expression (HEK-293T cells), strep-tagged human LIC1 (a.a. 1–354) was cloned into a pHR vector for lentiviral expression. Strep-tagged LIC1 was purified with Strep-Tactin resin (IBA, Germany) and eluted via a commercial Strep-Tactin elution buffer containing 2.5 mM desthiobiotin (Novagen, Germany). Human LIC1 was purified for RPLC with glutathione agarose 4B (USB) and then cleaved with GST-tagged human rhino-virus 3C protease overnight at 4°C. The cleaved product was further purified by gel filtration into 10 mM Tris (pH 7), 50 mM NaCl, 2 mM $MgCl_2$, and 2 mM TCEP.

FIP3 (human; AB383948.1), RILP (human; NM_031430.2), and BicD2 (mouse; NM_029791.4) were cloned into the vector pET28a, which encodes an N-terminal His tag with an additional N-terminal StrepII tag and superfolder-GFP. The proteins were expressed in BL21 DE3 RIPL cells overnight at 18°C and purified via Strep-Tactin resin (IBA). RILP and BicD2 were eluted via a commercial Strep-Tactin elution buffer (Novagen), and FIP3 was eluted using 2.5 mM d-Desthiobiotin (Sigma Aldrich) in 100 mM Hepes (pH 7.4), 10% glycerol, 0.5 mM EGTA, 5 mM $MgCl_2$, and 300 mM NaCl. RILP was puri-fied further by gel filtration into 50 mM Tris–HCl (pH 7), 150 mM NaCl, 2 mM $MgCl_2$, 1 mM EGTA, and 2 mM TCEP. FIP3 was further purified by gel filtration into 30 mM Hepes (pH 7.4), 50 mM K-Acetate, 2 mM Mg-Acetate, 1 mM EGTA, and 10% glycerol. GST-tagged superfolder GFP was purified via glu-tathione beads, followed by cleavage of the GST tag with human rhinovirus 3C protease. GFP was then further purified by gel filtration into 25 mM Hepes (pH 7.5), 150 mM NaCl, 10% glycerol, and 2 mM TCEP.

## Structure determination and refinement

Diffraction data were collected at the Advanced Light Source (ALS) (Lawrence Berkeley National Laboratory), beamline 8.3.1. Multi-wavelength anomalous dispersion (MAD) datasets were collected from selenomethionine (SeMet)-derivatized LIC crystals at two wavelengths, a high-energy remote wavelength and at a wavelength halfway between the peak and inflection point. A native data set was also collected from underivatized protein. The SeMet data sets were indexed to $P3_221$ and merged using HKL2000 (*Otwinowski and Minor, 1997*), and substructure determination (four selenium sites) and initial phases were obtained using AutoSol in Phenix (*Adams et al., 2010*). The structure was built using Coot (*Emsley and Cowtan, 2004*), and AutoBuild in Phenix (*Adams et al., 2010*) improved the initial model. After several rounds of refinement (via phenix.refine), the initial 3.6 Å structure was used as a search model for molecular replacement of the 2.1 Å native dataset using Phaser (*Adams et al., 2010*). The 2.1 Å native dataset was integrated and indexed to $C 2 2 2_1$ using XDS (*Kabsch, 2010*) and scaled and merged using XSCALE (*Kabsch, 2010*). After successful molecular replacement, several rounds of model building and refinement were carried out using Coot (*Emsley and Cowtan, 2004*) and phenix.refine (*Adams et al., 2010*). Final data collection and refinement statistics can be found in *Table 1*.

## Reverse-phase HPLC

The HPLC system (Waters) was used with a C18 column (Phenomenex, Torrance, CA). The solutions used were as follows: buffer A consisted of 5% acetonitrile, 5 mM tetrabutylammonium hydroxide, 25 mM $KH_2PO_4$ (pH 6); buffer B consisted of 60% acetonitrile, 5 mM tetrabutylammonium hydroxide, 25 mM $KH_2PO_4$ (pH 6.0). The gradient for an RPLC run was 0% to 65% buffer B over 44 min with an 18 μl

injection of sample. Nucleotides GMP, GDP, and GTP were purchased from Sigma Aldrich, and ppGpp was purchased from TriLink Biotechnologies (San Diego, CA). The synthetic nucleotide KTP was purchased from BIOLOG Life Science Institute (Germany) and used as a positive control for sample injection.

## Cell culture, transfection, and viral transduction

HEK-293T cells were cultured in DMEM media containing 10% FBS and 5% penicillin/streptomycin/glutamine. Mammalian expression LIC constructs were encoded with an mCherry reporter in the following construct: pHR-mCherry-p2A-LIC. The pHR construct was co-transfected with the plasmids pMD2.G and pCMVΔ8.91 for lentivirus production. To transfect cells, cells in one well of a six-well dish were transfected with 10 μl of 2 mg/ml polyethylenimine (Polysciences, Inc. Warrington, PA) and 2 μg of total DNA. The virus-containing media were collected after 3 days of incubation following transfection, and cell particulates were spun out. The virus was then concentrated 10-fold in PBS (Gibco, Grand Island, NY) using Lenti-X concentrator (Clontech, Mountain View, CA). To amplify protein expression, the lentivirus was used to infect HEK-293T cells for immunoprecipitations. Infections were conducted by adding 10–20 μl of the concentrated virus to a six-well dish of HEK-293T cells at ~50% confluency. The cells were then passaged to expand the cell quantity for immunoprecipitations.

## Protein interaction assays and Western blotting

### Immunoprecipitations

Following infection and expansion, HEK-293T cells were trypsinized, centrifuged, and resuspended with lysis buffer, which was composed of 50 mM Tris (pH 7), 100 mM NaCl, 1 mM EGTA, 1% Triton-X 100, and a protease inhibitor cocktail at 1 tablet/50 ml (complete mini-EDTA-free tablets from Roche, Indianapolis, IN). The lysate was incubated for 20 min on ice and then centrifuged at 20,000×$g$ for 10 min. The supernatant was used for immunoprecipitations (IPs). Protein G Dynabeads (Novex, Carlsbad, CA) were used with a magnetic bead separator (Invitrogen, Carlsbad, CA). For the IP, 50 μl of Dynabeads and 5 μg of primary antibody were incubated for 10 min at room temperature. The primary antibodies included rabbit anti-HA antibody (Rockland, Gilbertsville, PA) or mouse anti-HA antibody (Millipore, Billerica, MA). The beads were then washed with PBST once, followed by incubation with cell lysate at room temperature for 30 min. The Dynabeads were washed three times with 50 mM Tris (pH 7), 100 mM NaCl, and 0.1% Tween-20 and finally resuspended with 24 μl of 1× loading buffer. Equal volumes of samples were analyzed by SDS-PAGE.

### Pulldowns

A 300 μl volume of 200 nM full-length or truncated GST-tagged human LIC1 was incubated with 20 μl of glutathione resin for 1 hr at 4°C. The resin was washed with 50 mM Tris (pH 7), 100 mM NaCl, 0.1% Tween, and 1 mg/ml BSA. A 300-μl volume of 200 nM GFP-FIP3, GFP-RILP, or GFP-BicD2 was then incubated with GST-LIC1 for 1 hr at 4°C. The resin was finally washed and resuspended with 24 μl of 1× loading buffer. Equal volume of samples was analyzed by SDS-PAGE.

### Western blots

Samples were resolved on NuPAGE gels, which were transferred to nitrocellulose membranes using the iBlot Gel Transfer Device (Invitrogen). The blots were probed with primary antibody for 1 hr at room temperature, and the primary antibodies included rabbit anti-HA antibody (Rockland, 1:1000), mouse anti-HA (Millipore, 1:1000), mouse anti-dynein intermediate chain (clone 74.1, Millipore, 1:2000), rabbit anti-dynein heavy chain (clone KIAA0325; Proteintech, Chicago, IL, 1:500), and mouse anti-GFP (clone 3E6, Molecular Probes, Eugene, OR, 1:1000). The blots were then washed three times with TBST followed by incubation with 1:10,000 anti-mouse or anti-rabbit horseradish peroxidase-linked secondary antibody (GE Healthcare), anti-mouse-800 (Rockland), or anti-rabbit 680 (Molecular Probes) for 45 min at room temperature. Blots were developed either with Amersham ECL Western blotting detection reagent (GE Healthcare) or scanned using an Odyssey CLx Infrared Imaging System (LI-COR, Lincoln, NE).

## Phylogenetic analysis

Approximately, 182 sequences of members of the Ras superfamily were used to compare the LIC's placement within this protein family. Most sequences included those used in a phylogenetic study of the Ras superfamily (*Rojas et al., 2012*). Other sequences included the sequences of purported 'pseudo-GTPases', CheY, and CENP-M. In our analysis, a structure-based sequence alignment was made with *C. thermophilum* LIC and the Dali server's top hits for structural similarity to the

*C. thermophilum* LIC G domain (Rab33, Rab28, and Rab32). This alignment, made by the Dali server (*Holm and Rosenstrom, 2010*), was used as a profile, and the collection of Ras superfamily sequences was aligned to this profile using MafftWS (*Katoh and Standley, 2013*). The resulting alignment was modified by deleting segments in which less than 80% of the sequences did not align. The modified alignment was then used to generate a phylogenetic tree using Maximum Likelihood (*Guindon et al., 2010*). The ATGC PhyML was run using the LG substitution model, SPR and NNI tree improvement, 5 random starting trees, and 300 bootstraps. The resulting tree (*Figure 1—figure supplement 3*) was displayed using iTOL (*Letunic and Bork, 2007*).

## Mass spectrometry

Strep-tagged human LIC1 and porcine brain tubulin, which was purified as described in *Castoldi and Popov (2003)*, was buffer exchanged into 50 mM NH$_4$OAc and concentrated to approximately 0.2–4 µg/µl. The protein was then boiled for 8 min. The denatured protein was pelleted, and the supernatant was isolated for LC-MS analysis by the Vincent Coates Foundation Mass Spectrometry Laboratory (Stanford University). All LC-MS analyses were carried out by negative electrospray ionization using Agilent 1100 HPLC and linear ion trap mass spectrometer LTQ XL (Thermo Fisher Scientific, Waltham, MA). HPLC conditions: Merck ZIC-HILIC, PEEK Column, 100 mm × 2.1 mm ID, 3.5 µm, 200 Å; flow rate 0.3 ml/min. The gradient elution was from 70% to 10% (B) in 5 min. The mobile phase consisted of A: 90 mM ammonium acetate in water and B: acetonitrile/10 mM ammonium acetate buffer (9:1 vol/vol); the total run time was 9 min. Samples were diluted 4-fold with acetonitrile, and 10 µl of the sample was injected. The retention times were 2.32 min and 2.57 min for GDP and GTP, respectively. Ionization efficiency and the fragmentation pattern were evaluated by infusion of standard solutions of 10 µM GDP and GTP. Mass spectrometry method: three scan events were monitored. Scan 1: full scan with the mass range of 160–800 Da. Scan 2: CID (collision induced dissociation) of m/z = 442 Da (GDP). Scan 3: CID of m/z = 522 Da (GTP). LCquan software was used for data analysis. Fragment ions m/z = 344 Da (GDP) and m/z = 424 Da (GTP) were utilized for GDP and GTP detection. The analyte identification was performed based on the retention time of analytes and their fragmentation pattern comparing unknown and standard samples. After each sample or standard, a blank reagent was injected to minimize possible carryover.

## Accession numbers

Protein Data Bank Accession Code is 4W7G.

## Acknowledgements

We thank Damian Ekiert for his guidance with crystallographic data acquisition, data processing, and structure building. The staff at ALS beamline 8.3.1, especially James Holton, offered additional guidance in data acquisition for which we are grateful. We thank Danica Fujimori's laboratory members at UCSF, especially Idelisse Ortiz, for access to their HPLC and their helpful discussions in determining nucleotide identity. We thank Brooke Gardner and Elif Karagoz for discussions concerning *Chaetomium thermophilum* and providing cDNA. We would like to thank David King and Dave Maltby for their help in trying to use mass spectrometry for identification of ppGpp, and we would like to acknowledge Theresa McLaughlin and Ludmila Alexandrova of the Vincent Coates Foundation Mass Spectrometry Laboratory (Stanford University) for LC-MS analyses. We thank Richard McKenney for helpful discussions and for the GFP-RILP and GFP-BicD2 constructs. We also thank Walter Huynh for cloning and purifying GFP-FIP3, and Jacob Pfeil for purifying GFP.

## Additional information

### Funding

| Funder | Grant reference number | Author |
| --- | --- | --- |
| National Institutes of Health | R37GM038499 | Ronald D Vale |
| Howard Hughes Medical Institute | | Ronald D Vale |
| Genentech Foundation | Graduate Student Fellowship | Nicholas T Hertz |

| Funder | Grant reference number | Author |
|---|---|---|
| National Science Foundation | Graduate Student Fellowship | Courtney M Schroeder |
| National Institutes of Health | R01GM097312 | Ronald D Vale |
| National Institutes of Health | 5R01AI099243 | Jonathan ML Ostrem |

The funders had no role in study design, data collection and interpretation, or the decision to submit the work for publication.

## Author contributions

CMS, Conception and design, Acquisition of data, Analysis and interpretation of data, Drafting or revising the article; JMLO, Analysis and interpretation of data, Drafting or revising the article, Contributed unpublished essential data or reagents; NTH, Analysis and interpretation of data, Contributed unpublished essential data or reagents; RDV, Conception and design, Analysis and interpretation of data, Drafting or revising the article

# Additional files

## Major datasets

The following dataset was generated:

| Author(s) | Year | Dataset title | Dataset ID and/or URL | Database, license, and accessibility information |
|---|---|---|---|---|
| Schroeder CM, Ekiert DC, Vale RD | 2014 | Crystal Structure of the Dynein Light Intermediate Chain's Conserved Domain | http://www.pdb.org/pdb/explore/explore.do?structureId=4w7g | Publicly available at the Protein Data Bank (http://www.rcsb.org/pdb/). |

The following previously published datasets were used:

| Author(s) | Year | Dataset title | Dataset ID and/or URL | Database, license, and accessibility information |
|---|---|---|---|---|
| Pai EF, Krengel U, Petsko GA, Goody RS, Kabsch W, Wittinghofer A | 1990 | Refined crystal structure of the triphosphate conformation of h-ras p21 at 1.35 angstroms resolution: implications for the mechanism of gtp hydrolysis | http://www.rcsb.org/pdb/explore/explore.do?structureId=5p21 | Publicly available at the Protein Data Bank (http://www.rcsb.org/pdb/). |
| Pan X, Eathiraj S, Munson M, Lambright DG | 2006 | Crystal Structure of Gyp1 TBC domain in complex with Rab33 GTPase bound to GDP and AlF3 | http://www.rcsb.org/pdb/explore/explore.do?structureId=2G77 | Publicly available at the Protein Data Bank (http://www.rcsb.org/pdb/). |
| Lee SH, Baek K, Dominguez R | 2008 | Crystal Structure of Rab28 GTPase in the Active (GppNHp-bound) Form | http://www.rcsb.org/pdb/explore/explore.do?structureId=3e5h | Publicly available at the Protein Data Bank (http://www.rcsb.org/pdb/). |
| Hesketh GG, Perez-Dorado I, Jackson LP, Wartosch L, Schefer IB, Gray SR, Mccoy AJ, Zeldin OB, Garman EF, Harbour ME, Evans PR, Seaman MN, Luzio JP, Owen DJ | 2014 | Complex of human VARP-ANKRD1 with Rab32-GppCp | http://www.rcsb.org/pdb/explore/explore.do?structureId=4CYM | Publicly available at the Protein Data Bank (http://www.rcsb.org/pdb/). |
| Wang ZX, Shi L, Liu JF, An XM, Chang WR, Liang DC | 2005 | Crystal structure of human ARL5 | http://www.rcsb.org/pdb/explore/explore.do?structureId=1zj6 | Publicly available at the Protein Data Bank (http://www.rcsb.org/pdb/). |
| Buglino J, Shen V, Hakimian P, Lima CD | 2002 | Structure of the Obg GTP-binding protein | http://www.rcsb.org/pdb/explore/explore.do?structureId=1LNZ | Publicly available at the Protein Data Bank (http://www.rcsb.org/pdb/). |

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
