## [Decision Letter]

Thank you for sending your work entitled “A ras-like domain in the light intermediate chain bridges the dynein motor to a cargo-binding region” for consideration at *eLife.* Your article has been favorably evaluated by K (Vijay) VijayRaghavan (Senior editor) and 3 reviewers, one of whom is a member of our Board of Reviewing Editors. We believe that your data showing that LIC is a bipartite protein containing an N-terminal G-domain that mediates the interaction with Dynein, and a C-terminal region linking Dynein to Rab effectors is an important step forward in the field.

The Reviewing editor and the other reviewers discussed their comments before we reached this decision, and the Reviewing editor has assembled the following comments to help you prepare a revised submission.

There were two major points that were brought up.

1) First there is the issue of to what extent the fold can simply be identified by bioinformatics, and the fact that the paper seems to gloss over this point. Many proteins, including proteins in bacteria (e.g. CheY), share the fold of Ras-type GTPases but do not bind or hydrolyze GTP and do not bear sufficient sequence identify with GTPase to allow discerning evolutionary relatedness. But the concept is per se not novel. What is missing in this manuscript is a more detailed analysis of the position of the LIC GTPase in the Ras superfamily. It is important to show a structure-based alignment of the LIC sequence with Ras and other selected members of the Ras family. What level of sequence identity is present? Is it possible to hypothesize, based on sequence identity/similarity, from which branch of the Ras superfamily has the LIC domain originated? Similarity of the DLIC sequence with small Ras-family GTPases can be quite easily evinced from browsing the NCBI web site, where family pfam05783 (LIC) is clearly indicated as a member of the superfamily cl17170 (Ras superfamily). This relationship can be easily identified by BLASTing segments of the sequence of LIC. This does not diminish the importance of the work, but we feel that you should give a fair representation of the state of the art and acknowledge that the relationship has been simply overlooked until now, and is not being revealed here for the first time though.

2) Secondly, we had a concern about whether the LIC G domain picks up in bacteria is not ppGpp. Whatever it is that the protein binds to, it elutes closely but not exactly where the ppGpp standard elutes, whereas if you look at other nucleotides the overlap is perfect. For instance we compared Figure 4. 4D is particularly revealing because it shows that if you mix ppGpp to the LIC G domain it elutes exactly with the standard, whereas the nucleotide bound to the LIC1 G domain purified from bacteria elutes a bit earlier. Same in panel F, it is a reproducible difference. So in essence we don't think that the authors can conclude from this figure that ppGpp is the better binder, because there is no direct comparison. We are worried that for your current model to hold up, you need to prove that ppGpp is indeed a better binder than GDP. However this depends to a certain extend on whether ppGpp is present in human cells or specific to bacteria. You could also express it using baculoviral expression (or better yet human cells) and confirm whether or not GDP nucleotide is bound.

---

## [Author Response]

*1) First there is the issue of to what extent the fold can simply be identified by bioinformatics, and the fact that the paper seems to gloss over this point. Many proteins, including proteins in bacteria (e.g. CheY), share the fold of Ras-type GTPases but do not bind or hydrolyze GTP and do not bear sufficient sequence identify with GTPase to allow discerning evolutionary relatedness. But the concept is per se not novel. What is missing in this manuscript is a more detailed analysis of the position of the LIC GTPase in the Ras superfamily. It is important to show a structure-based alignment of the LIC sequence with Ras and other selected members of the Ras family. What level of sequence identity is present? Is it possible to hypothesize, based on sequence identity/similarity, from which branch of the Ras superfamily has the LIC domain originated? Similarity of the DLIC sequence with small Ras-family GTPases can be quite easily evinced from browsing the NCBI web site, where family pfam05783 (LIC) is clearly indicated as a member of the superfamily cl17170 (Ras superfamily). This relationship can be easily identified by BLASTing segments of the sequence of LIC. This does not diminish the importance of the work, but we feel that you should give a fair representation of the state of the art and acknowledge that the relationship has been simply overlooked until now, and is not being revealed here for the first time though*.

While not discussed in prior publications, we agree that the NCBI web site identifies the LIC as a member of the Ras superfamily. Thus, we have now acknowledged the Pfam database’s classification of the LIC in both the Introduction and the Results section. Furthermore, we have pointed out in the Discussion that CheY and CENP-M (recently published in *eLife*) also have Ras-like folds without nucleotide binding capabilities, thus revealing the presence of other “pseudo-GTPases”.

The referees also requested a more detailed analysis of the LIC’s position in the Ras superfamily. We used the Dali server (23) to compare the *C. thermophilum* LIC structure to all other structures in the PDB. The structure of the LIC was found to be most similar to certain Rab proteins, specifically Rab33, Rab32 and Rab28. Despite the structural similarity, the sequence identity is quite low (approximately 10%). The structural and sequence alignments with the top three Dali server hits are displayed in Figure 1—figure supplement 2 and are described in the Results section.

To determine the LIC’s placement in the Ras superfamily, we have conducted a similar phylogenetic analysis to that described in the recent *eLife* paper “The pseudo GTPase CENP-M drives human kinetochore assembly” (7). We first performed a structure-based alignment of fungal and metazoan LICs to 167 Ras superfamily representatives used in a bioinformatics study (51). We also included the sequences of pseudo-GTPases CheY and CENP-M from a number of species in the alignment. This alignment was used to generate a maximum likelihood phylogenetic tree with 300 bootstraps to determine clade probability. The tree shows that the LICs may share common ancestry with the Ras and Rab subfamilies, as was found with CENP-M (7). We have displayed the phylogenetic tree in Figure 1—figure supplement 3, and it is described in the Results section.

*2) Secondly, we had a concern about whether the LIC G domain picks up in bacteria is not ppGpp. Whatever it is that the protein binds to, it elutes closely but not exactly where the ppGpp standard elutes, whereas if you look at other nucleotides the overlap is perfect. For instance we compared*
Figure 4*. 4D is particularly revealing because it shows that if you mix ppGpp to the LIC G domain it elutes exactly with the standard, whereas the nucleotide bound to the LIC1 G domain purified from bacteria elutes a bit earlier. Same in panel F, it is a reproducible difference. So in essence we don't think that the authors can conclude from this figure that ppGpp is the better binder, because there is no direct comparison. We are worried that for your current model to hold up, you need to prove that ppGpp is indeed a better binder than GDP. However this depends to a certain extend on whether ppGpp is present in human cells or specific to bacteria. You could also express it using baculoviral expression (or better yet human cells) and confirm whether or not GDP nucleotide is bound*.

The major comment here was ascertaining whether GDP is indeed the nucleotide bound to human LIC when this protein is expressed in human cells (since we used human LIC, we agreed that human cells would be the ideal choice for this experiment). We should add that ppGpp is thought to be specific to bacteria and plants (59; 62) and not human cells. In this revision, we expressed the human LIC1 G domain in human HEK-293T cells; the purified G domain was denatured by boiling, and the supernatant was analyzed by LC-MS. The mass spec clearly detected GDP with little or no GTP (see Figure 4—figure supplement 3). Therefore, GDP is indeed the primary nucleotide bound to human LIC1 in its native environment.

This comment also states that our model claims that ppGpp is a “better binder” than GDP. However, we do not claim that ppGpp is a better binder than GDP (our model states that LIC binds GDP preferentially over GTP, which is supported by the data in the paper). We believe that ppGpp most likely binds with equal affinity as GDP. In the crystal structure of Arl5 with bound ppGp (a remnant of ppGpp), only the one diphosphate is bound in the active site, while the other phosphate is solvent exposed and not making contact with the protein (see structure below; (68)). We speculate that the same situation applies to LIC1, where one of the diphosphates of ppGpp is bound to LIC1 (like GDP), while the other diphosphate extends into the solvent. We have now clarified these points.

*Crystal structure of Arl5*-*GDP3'P*

GDP3'P, a likely remnant of ppGpp, makes contacts with residues of Arl5 with only one diphosphate, analogous to GDP (PDB: 1ZJ6) (68).Author response image 1.

So why does human LIC1 co-purify with ppGpp when expressed in bacteria but GDP when expressed in human cells? When bacteria are stressed with high protein expression, ppGpp can reach millimolar quantities (59), indeed much higher than the approximate 200 µM GDP in exponential growth (9). As described earlier, ppGpp is an alarmone that is unique to bacteria and plants (59; 62) and not found in animal cells. Thus, we would like to emphasize that the co-purification of ppGpp with human LIC1 is simply an artifact of its purification from bacteria. It is also important to note that approximately 2.5 mg/mL of protein at a minimum is required to detect nucleotide above the noise in the RPLC method described, and thus GDP may also co-purify with bacterially expressed human LIC1, though in quantities below our detection limit. Nonetheless, the finding of ppGpp gave us the important clue that human LIC1 binds guanosine diphosphates preferentially over guanosine triphosphates, which we demonstrate with purified protein in Figure 4. We thank the referees for expressing their concerns, and we have made these points clearer.

The referees also raised the point that ppGpp might not be the nucleotide co-eluting with bacterially expressed human LIC1 due to the slight mismatch of the retention time with commercial ppGpp. The referees raise a valid point that the retention times of the LIC1 nucleotide shown in Figure 4 are not exactly the same as commercial ppGpp, whereas the commercial GDP in Figure 4 appears exactly the same between two HPLC runs. However the retention times of nucleotides do vary among HPLC runs, as was also noted in an earlier study using the same assay (22).

In this revised manuscript, we quantified the retention time of all standards and the LIC nucleotide for more than 3 runs and display the standard deviation as error bars in Figure 4—figure supplement 1. The retention time of LIC1’s nucleotide has a higher standard deviation (±0.355 min) than the commercial ppGpp (±0.138 min), yet this variability is less than seen with even GMP, GDP and GTP (±0.893, 0.494, and 0.643 min, respectively). Furthermore, the difference in retention times seen for the LIC nucleotide and ppGpp is not significantly different (p = ∼0.2). Furthermore, the retention time of the LIC nucleotide shows that it is more electronegative than GTP, making ppGpp a good candidate. Of course, mass spectrometry would be more definitive, but ppGpp is technically very difficult to identify by mass spectrometry. In the revised manuscript, we now state, “…with indistinguishable RPLC elution times and a guanine-like absorption spectrum, we believe that ppGpp is the most likely candidate for the LIC-bound nucleotide.” Furthermore, since we now show that GDP binds the LIC when expressed in human cells, it also becomes less essential to have complete certainty of the bacterially expressed nucleotide. The overall data (from bacterial expression, human cell expression and exchange studies with purified nucleotide) collectively suggest that the LIC has a preference for binding a diphosphate rather than triphosphate guanine nucleotides.